# Trends in Incidence and Mortality of Head and Neck Cancer Subsites Among Elderly Patients: A Population-Based Analysis

**DOI:** 10.3390/cancers17030548

**Published:** 2025-02-06

**Authors:** Małgorzata Wierzbicka, Wioletta Pietruszewska, Adam Maciejczyk, Jarosław Markowski

**Affiliations:** 1Department of Otolaryngology, Regional Specialist Hospital Wroclaw, Research & Development Centre, 51-124 Wroclaw, Poland; wierzbicka.otolaryngology@gmail.com; 2Faculty of Medicine, Wroclaw University of Science and Technology, 50-370 Wroclaw, Poland; 3Institute of Human Genetics, Polish Academy of Sciences, 60-479 Poznan, Poland; 4Department of Otolaryngology Head Neck Oncology, Medical University of Lodz, 90-419 Lodz, Poland; 5Department of Oncology, Wroclaw Medical University, 53-413 Wroclaw, Poland; adam.maciejczyk@dcopih.pl; 6Department of Laryngology, Faculty of Medical Sciences in Katowice, Medical University of Silesia in Katowice, 40-027 Katowice, Poland; jmarkowski@sum.edu.pl

**Keywords:** head and neck cancer (HNC), incidence and mortality trends, elderly population, HPV-related cancers, gender differences, cancer prevention strategies

## Abstract

The incidence of head and neck cancer (HNC) has been rising over the past two decades, especially among elderly patients. Using data from the Polish Cancer Register (1999–2021), this study analyzed trends in HNC incidence and mortality across three age groups: 60–69, 70–79, and 80+. The results showed that incidence increased faster than mortality in most groups, particularly for oral and oropharyngeal cancers in women. In patients aged 80+, mortality rates rose significantly for certain cancers, including laryngeal and oral cancers in men. Women experienced the sharpest increases in oral cancer across all age groups. These findings emphasize the need for targeted prevention, such as HPV vaccination, and improved early detection strategies for older adults, particularly for high-risk groups like older women.

## 1. Introduction

Accurate epidemiological data are essential for assessing the care needs of the growing population of head and neck cancer (HNC) patients. HNCs are the sixth most common cancers worldwide, with a consistently increasing overall incidence and a predicted 30% annual rise by 2030 [1,2,3,4]. HNCs affect men two to four times more frequently than women, with estimates exceeding 20 per 100,000 [2,5,6]. While laryngeal cancer is often reported as the most common head and neck cancer subtype globally, with a 23% increase in the past decade, some databases indicate higher incidence rates for oral cancers, particularly in certain regions. However, age-adjusted incidence rates for laryngeal cancer have been declining in countries with higher sociodemographic indices, reflecting shifts in smoking and alcohol consumption patterns [5]. In Europe, laryngeal cancer accounts for 2–5% of all cancers, with higher incidence rates in males compared to females [7]. Nevertheless, recent years have seen a rise in cases among younger women, particularly in European countries, potentially attributable to gender-specific trends in tobacco and alcohol use [8,9]. In a U.S. cohort study from 1986 to 2018, the incidence of laryngeal cancer decreased, although the corresponding decline in mortality was not as significant. Oral and lip cancers are the second most common cancers in men in developing countries, with an incidence of 10 per 100,000 [10]. The incidence of oral cancer is expected to increase across Southeast Asia in line with population growth [11]. The incidence of oral and oropharyngeal cancers in males has decreased in countries such as France (−12.6%), Slovakia (−4.0%), and Spain (−10.8%), while increases have been observed in the UK (18.8%), Australia (8.7%), Japan (21.3%), and the USA (3.7%) [8].

However, the worldwide incidence of HNC is influenced not only by a complex interplay of epidemiological factors but also by population aging, particularly in highly industrialized nations. The risk of HNC increases with age across populations, with most cases diagnosed in individuals over the age of 50. Knowledge gaps and future directions for the assessment and treatment of elderly HNC patients were identified by the CARG-HNC Study Group in 2017 [12]. One challenge is the need to monitor the scale of the problem, i.e., tracking trends in HNC incidence among older adults, as well as analyzing the specificity, primary location, and treatment outcomes by age. Key questions include how the incidence of specific HNC subsites changes with age and gender, as well as how treatment outcomes are reflected in the deviations between parallel trends in morbidity and mortality. Consequently, the aim of this study was to conduct a detailed analysis of three senior age cohorts (60–69, 70–79, and 80+) over the 22-year period from 1999 to 2021, stratified by HNC subsite, gender, and age, and to project incidence and mortality trends through to 2035.

## 2. Materials and Methods

### 2.1. Analysis of HNC by Primary Location, Gender, and Age Cohorts

Different primary locations of HNC were analyzed by gender, age cohort, and time in elderly Polish patients, with trend simulations included. The data source was the population-based Polish National Cancer Register, which is published periodically (https://onkologia.org.pl/en/ (accessed on 1 December 2024)). HNC subsites were classified according to the International Classification of Oncology (ICD-10) codes for malignant neoplasms of the lip (C00), the tongue (C01), the other and unspecified parts of the tongue (C02), the gym (C03), the floor of the mouth (C04), the palate (C05), the other and unspecified parts of the mouth (C06), the parotid gland (C07), the major salivary glands (C08), the tonsil (C09), the oropharynx (C10), the nasopharynx (C11), the pyriform sinus (C12), the hypopharynx (C13), the nasal cavity and middle ear (C30), the accessory sinuses (C31), and the larynx (C32) from the World Health Organization. Thyroid, esophageal, and nasopharyngeal cancers were excluded due to their low incidence in the Polish population and differences in etiology and management compared to other HNC subsites (https://onkologia.org.pl/en/ (accessed on 1 December 2024)).

The study population consisted of 85,032 patients, including 19,198 females and 65,834 males. The age cohorts of 60–69, 70–79, and 80+ years included 9335, 6145, and 3718 females and 39,928, 19,884, and 6022 males, respectively. The total number of deaths was 59,386, comprising 12,440 females and 46,964 males. For the 60–69, 70–79, and 80+ cohorts, there were 4846, 3857, and 2737 female deaths and 26,157, 14,845, and 5944 male deaths, respectively (Figure 1).

Given the differences in presenting symptoms, treatment regimens, and prognosis for each anatomical subsite, the primary locations of HNC were grouped into seven predominant anatomical sites for analysis: larynx, lip, oral cavity, mesopharynx, hypopharynx, major salivary glands, and paranasal sinuses. Thyroid and esophageal cancers were excluded, as they are typically omitted in epidemiological studies. Additionally, nasopharyngeal cancer was excluded due to its rarity in the Polish population.

The analysis for the 60–69, 70–79, and 80+ age cohorts covered cancer incidence and mortality in individual locations for women, men, and the general population during the years 1999–2021. The primary focus was on the results of parallelism tests between the trend lines for incidence and mortality in the analyzed primaries and age cohorts.

### 2.2. Statistical Methods

Regression and correlation analyses were conducted on incidence and mortality data for the years 1999–2021 across the 60–69, 70–79, and 80+ age cohorts. The significance of correlations was assessed using *t*-tests (α = 0.05, two-sided). Differences between regression slopes were evaluated using F-tests and pairwise comparisons with Bonferroni correction (αc ≈ 0.017). Relative incidence and mortality rates (using 1999 as the base year) were calculated to account for cohort size differences. Trend lines were extrapolated to 2035. Gender-specific comparisons of trend parallelism for HNC neoplasms were performed using F-tests and pairwise comparisons (αc = 0.00238). Cohort-specific slope differences were analyzed for each primary site. Incidence vs. mortality trend slopes were compared using parallelism tests (α = 0.05, one-sided) for all cancer sites, stratified by cohort and gender. The results are presented graphically and in tables.

## 3. Results

The trend lines of the absolute number of cases for the seven HNC locations, for the age cohorts 80+, 70–79, and 60–79, are shown in Figure 2, Figure 3 and Figure 4, labeled as a, b, and c, respectively.

### 3.1. Age Cohort 80+

Figure 1 illustrates the trend lines for the 80+ age cohort. In women, all seven trend lines had slopes significantly different from zero based on the *t*-test results for the correlation coefficient. The F-test results indicated that the slopes were significantly different (*p* < 0.001). Pairwise comparisons of the slopes of these trend lines, with Bonferroni correction applied, showed that six lines could be considered parallel. The exception was the trend line for oral cavity cancer, which exhibited the strongest and most significant upward trend (*p* < 0.001).

In men, the trends for lip and paranasal sinus cancers were not statistically significant, suggesting that the number of cases remained stable during the observation period. The trends for the remaining subsites were statistically significant and increasing. According to the F-test, the slopes of the trend lines were significantly different (*p* < 0.001), with laryngeal cancer showing the steepest slope, indicating a strong upward trend.

To summarize, in women, all seven HNC locations exhibited significant increasing trends, with oral cavity cancer showing the strongest growth. In men, the incidence of lip and paranasal sinus cancers remained stable, while that of other locations increased significantly, with laryngeal cancer showing the most pronounced growth (*p* < 0.001).

### 3.2. Age Cohort 70–79

In females, the incidence trends were predominantly increasing, except for lip cancer. In males, the trends varied: laryngeal and lip cancers showed a decreasing trend, while most other locations exhibited increasing trends, and paranasal sinus cancer remained stable. The combined trends mirrored the male patterns due to the higher prevalence of HNC in men.

### 3.3. Age Cohort 60–69

In females, the incidence trends increased across all locations except for lip cancer. In males, similar patterns were observed, with significant increases in all areas except for lip cancer, which showed a decreasing trend. The combined trends reflected these patterns, emphasizing the male predominance in HNC cases.

### 3.4. Incidence Trends by HNC Location

#### 3.4.1. Laryngeal Cancer

In females, the trend line slope for the 60–69 cohort was significantly steeper compared to the slopes for the 70–79 and 80+ cohorts (*p* < 0.05). No significant difference in slopes was observed between the 70–79 and 80+ cohorts (*p* = 0.09). In males and the combined analysis, the 60–69 cohort exhibited a significantly steeper slope than the 70–79 cohort. The 70–79 cohort had a decreasing trend, while the 60–69 cohort showed a significant increase in incidence over the observation period. The 80+ cohort displayed a rising trend like the 60–69 cohort. (Figure 5 and Figure 6).

#### 3.4.2. Lip Cancer

The trend line for the 80+ cohort had a significantly steeper positive slope compared to the other two cohorts (*p* < 0.001). The 60–69 and 70–79 cohorts exhibited negative slopes, indicating a decrease in the incidence of lip cancer in the two younger cohorts (*p* = 0.02), while the incidence increased in the 80+ cohort.

#### 3.4.3. Hypopharyngeal, Oropharyngeal, and Oral Cavity Cancers

All analyzed trend lines had positive slopes, indicating increasing trends, regardless of patient group and age cohort (*p* < 0.001). In each case, the 60–69 cohort had the steepest slopes, followed by the 70–79 cohort, while the 80+ cohort had the smallest slopes. All pairwise comparisons of trend lines yielded *p*-values well below 0.017, confirming the significant difference in growth rates across cohorts.

#### 3.4.4. Salivary Gland Cancer

Increasing trends were observed across all groups, with the steepest increase in the 60–69 cohort, followed by the 80+ cohort, and the slowest increase in the 70–79 cohort. Significant differences in trend line slopes were observed, particularly between the 70–79 and 80+ cohorts in the male group (*p* = 0.04), where the incidence increased faster in the 80+ cohort.

#### 3.4.5. Paranasal Sinus Cancer

Positive slopes were observed for all cohorts, with the 60–69 cohort showing the steepest increase. The 70–79 and 80+ cohorts had very similar slopes. Statistical significance was found in the comparison between the 60–69 cohort and both the 70–79 and 80+ cohorts, with the youngest cohort showing a much greater increase in cases (*p* < 0.05). (Table 1 and Table 2).

## 4. Discussion

The authors aimed to compare the incidence and mortality trends between age cohorts and anatomical HNC subsites, indirectly allowing for the visualization of treatment effectiveness by age and HNC primaries. The researchers focused particularly on elderly patients, defined as those aged 60 and above, with comparisons made between patients in their seventh and eighth decades of life and the senior group aged 80 and above. Our findings confirm a continuous rise in HNC incidence since 1999, and projections suggest a sustained upward trend through to 2035, consistent with previously reported data [3,4]. Data on lip, oral cavity, and pharyngeal cancers extracted from the Cancer Incidence in Five Continents (CI5) volumes I-XI, the Nordic Cancer Registries (NORDCAN), the Surveillance, Epidemiology, and End Results (SEER) Program, and the WHO IARC mortality database revealed that, although global trends showed an overall decrease in incidence and mortality, significant increases were observed in older age groups and among female subjects [13]. The gender gap was more than twice as large, favoring males, while the trend lines for females demonstrated a much steeper increase. The risk of HNC increases with age across populations, with most cases diagnosed in individuals over 50 years old [2,14]. Notably, a rise in cases among younger women has been observed, particularly in European countries, which may be explained by gender-specific patterns of tobacco and alcohol consumption [9]. Among elderly patients, laryngeal and oral cavity cancers were the most common, while nasal cavity and nasopharyngeal cancers were the rarest [14].

Globally, laryngeal cancer remains the most common HNC subtype, with a 23% increase over the past decade. However, age-adjusted incidence rates for laryngeal cancer have been declining in countries with a higher sociodemographic index, reflecting changes in smoking and alcohol consumption behaviors. In Europe, laryngeal cancer accounts for 2–5% of all cancers, with a much higher incidence among males than females [7]. In a U.S. cohort study from 1986 to 2018, the incidence of laryngeal cancer decreased, although mortality rates did not decline at the same rate. In our study, we observed negative trends for laryngeal cancer in men aged 60–80; however, there was a marked increase in incidence among those aged 80 and above. Conversely, among women, the number of cases showed a statistically significant rise.

Lip and oral cavity cancers are the second most common cancers among men in developing countries (10 per 100,000) [10], and the incidence of oral cancer is expected to rise in Southeast Asia in line with population growth [11]. A key finding from the updated analysis of oral cancer mortality in 2011 was the leveling of the epidemic among men in most European countries, including Hungary and other Central European countries, where mortality from this cancer had been exceedingly high [15], as well as in Ireland. Our study confirms a substantial increase in oral cavity cancers, particularly among women, across all analyzed age cohorts. The incidence of lip cancer, however, is declining in nearly the entire analyzed population, except for the 80+ cohort.

The increasing incidence of HNC in the USA and Europe has been linked to a rise in oropharyngeal cancer, driven by human papillomavirus (HPV) infection [16,17]. Recent studies have demonstrated a global trend of increasing incidence in HPV-related subsites, while HPV-unrelated subsites have shown a decline in countries such as the USA, Canada, Hong Kong, and Korea [17]. Recent epidemiological data indicate varying trends in the incidence of oral and oropharyngeal cancer across different countries. Declines have been observed in France (−12.6%), Slovakia (−4.0%), Spain (−10.8%), Brazil (−26.7%), and Hong Kong (−10.5%), whereas increases have been noted in the UK (18.8%), Australia (8.7%), Japan (21.3%), and the USA (3.7%). The rising incidence of oropharyngeal cancer, particularly among elderly populations [18], has been linked to the increasing prevalence of HPV infections. While the relationship between HPV and oropharyngeal cancer is well established, its association with oral cancer remains a topic of ongoing investigation. Studies have detected HPV in salivary and biopsy samples from patients with oral cancer, highlighting a potential but not yet fully proven link. Further research is required to determine the extent of HPV’s role in oral cancer development [19].

In our analysis, we confirmed these findings, with a rapid increase in the incidence of oropharyngeal cancer in the 60–69 cohort, a slower rise in the 70–79 cohort, and the slowest increase in the 80+ cohort. This study’s findings regarding the rapid rise in oropharyngeal cancer, particularly in the 60–69 cohort, align with global data on the increasing prevalence of HPV-related cancers. This highlights the need for expanded HPV vaccination efforts, particularly in older age cohorts, as part of a broader cancer prevention strategy. Additionally, the slower rise in incidence in the 80+ cohort may reflect a generational lag in HPV exposure, suggesting that future generations of elderly patients could face higher incidence rates. This supports ongoing efforts to study and monitor HPV-related cancer trends among different age cohorts over time [20,21]. While HPV vaccination is well established for younger individuals, its utility in older populations remains uncertain and requires further investigation. Our findings underscore the need to explore the potential for expanding vaccination strategies to include older adults who may still benefit from protection against HPV-related cancers.

Additionally, a significant increase in the incidence of salivary gland cancer was observed, particularly in the 60–69 and 80+ cohorts, pointing to an emerging trend that may require further investigation. The reasons behind this rise are unclear and warrant further research, particularly to examine environmental or genetic factors that could be influencing this trend. As this cancer type is relatively rare, increased surveillance and better understanding of risk factors could lead to earlier detection and potentially improved outcomes.

Survival rates in the elderly European population were lower compared to younger individuals for all HNC types, ranging from approximately 60% for salivary gland and laryngeal cancers to 22% for hypopharyngeal tumors [14]. However, when excluding deaths within one year of diagnosis (likely advanced-stage disease), 5-year survival rates for the elderly exceeded 60%. The differences between age groups diminished and, in some cases, parity was achieved for laryngeal and oral cavity cancers [14]. While routine health checks for older patients encompass multiple medical concerns, the rising burden of head and neck cancers, particularly in high-risk subgroups, highlights the importance of integrating focused screening and preventive efforts within general geriatric care.

To address these trends, incidence and mortality trend lines were compared. In the 60–69 cohort, the incidence rate increased faster than the mortality rate. Similarly, in the 70–79 cohort, mortality increased at a slower rate than incidence, particularly in women with oropharyngeal and oral cavity cancers, and in both genders for salivary gland cancers. However, a different pattern emerged for the oldest patients. In the 80+ cohort, both slopes were positive, but the incidence rate increased more slowly than the mortality rate for all primary cancers in men, as well as for laryngeal, hypopharyngeal, and oral cavity cancers in the general population. While individual management decisions remain unaffected by these trends, clinicians may benefit from awareness of demographic and disease-specific patterns to better advocate for early detection resources and preventive interventions in high-risk populations.

There are some limitations of this study, like its retrospective nature and data sources or the lack of granular data on risk factors such as smoking history, alcohol consumption, HPV status, and socioeconomic factors. These variables can play a significant role in explaining regional and gender differences in HNC incidence and mortality, but their absence limits the ability to explore causal relationships and risk factor modification strategies. The exclusion of certain cancers, such as thyroid and esophageal cancers, as well as nasopharyngeal cancer due to its rarity in the Polish population, could also limit this study. While this exclusion is justified from an epidemiological standpoint, it may limit the comprehensiveness of this study in capturing the full burden of head and neck cancers in the elderly population. The limited scope of treatment modalities and outcomes may suggest indirect inferences about treatment effectiveness, and the absence of data on treatment types, adherence, and treatment-related factors limits the ability to draw firm conclusions about the impact of evolving therapies on patient outcomes, particularly in elderly cohorts who may face treatment challenges due to comorbidities. Although this study’s findings do not directly alter individual management, understanding these trends can inform resource allocation, targeted screening programs, and public health campaigns aimed at reducing the burden of head and neck cancers in aging populations.

## 5. Conclusions

This analysis reveals distinct trends in HNC incidence across sex and age cohorts from 1999 to 2021. The elderly population is not homogeneous in terms of HNC incidence and survival. Differences were observed between the 60–80 and 80+ age groups. While males and the total population exhibited declining trends across most age groups, females displayed a slight increase in the 80+ age cohort. These findings highlight the importance of age- and sex-specific strategies in HNC prevention and management.

Tailored strategies based on age and gender:This study highlights the need for prevention, early detection, and treatment strategies tailored to age and gender. While younger cohorts show more favorable trends, the oldest patients (80+) face higher mortality rates. Special attention should be given to the rising incidence among women, especially in oral cavity cancers.HPV vaccination and public health efforts:The increasing incidence of HPV-related oropharyngeal cancer supports expanding HPV vaccination programs. Prevention efforts should target not only younger populations but also older adults who may still benefit from vaccination and screening programs.Importance of early detection and intervention:The observed differences in survival rates highlight the critical role of early diagnosis in improving HNC outcomes. Efforts should focus on raising awareness, enhancing screening, and ensuring timely diagnosis for high-risk populations.Future research needs:Further research is needed to understand the rising incidence of salivary gland cancer and explore interventions for elderly populations. Investigating environmental and genetic factors contributing to HNC’s incidence could help tailor prevention strategies more effectively.

## Figures and Tables

**Figure 1 cancers-17-00548-f001:**
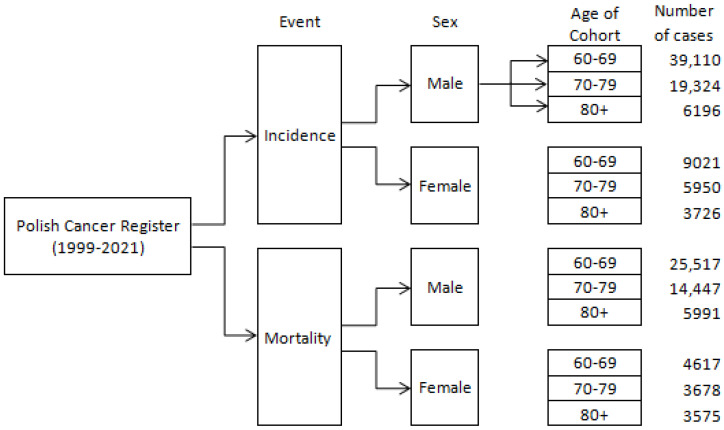
Flowchart describing study group.

**Figure 2 cancers-17-00548-f002:**
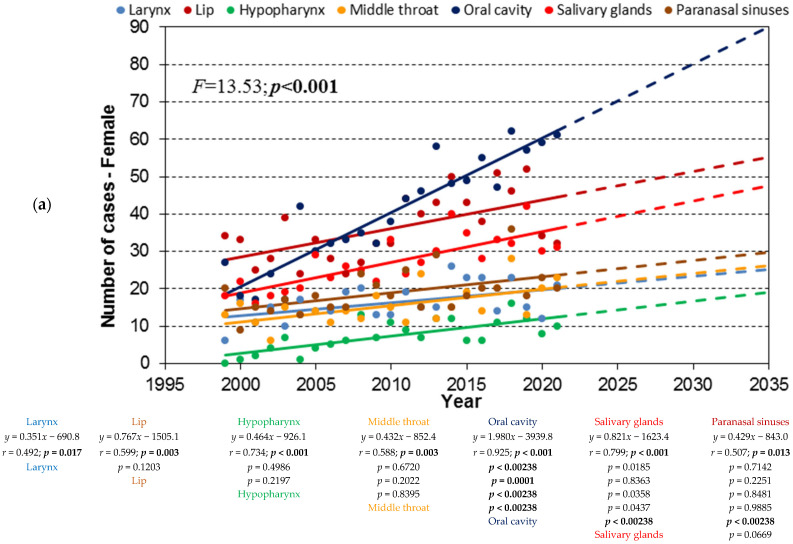
(**a**). Trend lines of the absolute number of cases of women with distinguished groups of HNSCC for the age cohort 80+; dashed lines show the extrapolation of the trend to 2035 determined in the period 1999 − 2021. Head and neck squamous-cell carcinoma (HNSCC). (**b**). Trend lines of the absolute number of cases of men with distinguished groups of HNSCC cancers for the age cohort 80+; dashed lines show the extrapolation of the trend to 2035 determined in the period 1999 − 2021. (**c**). Trend lines of the absolute number of cases of distinguished groups of HNCs for the 80+ age cohort; dashed lines show the extrapolation of the trend to 2035 determined in the period 1999 − 2021.

**Figure 3 cancers-17-00548-f003:**
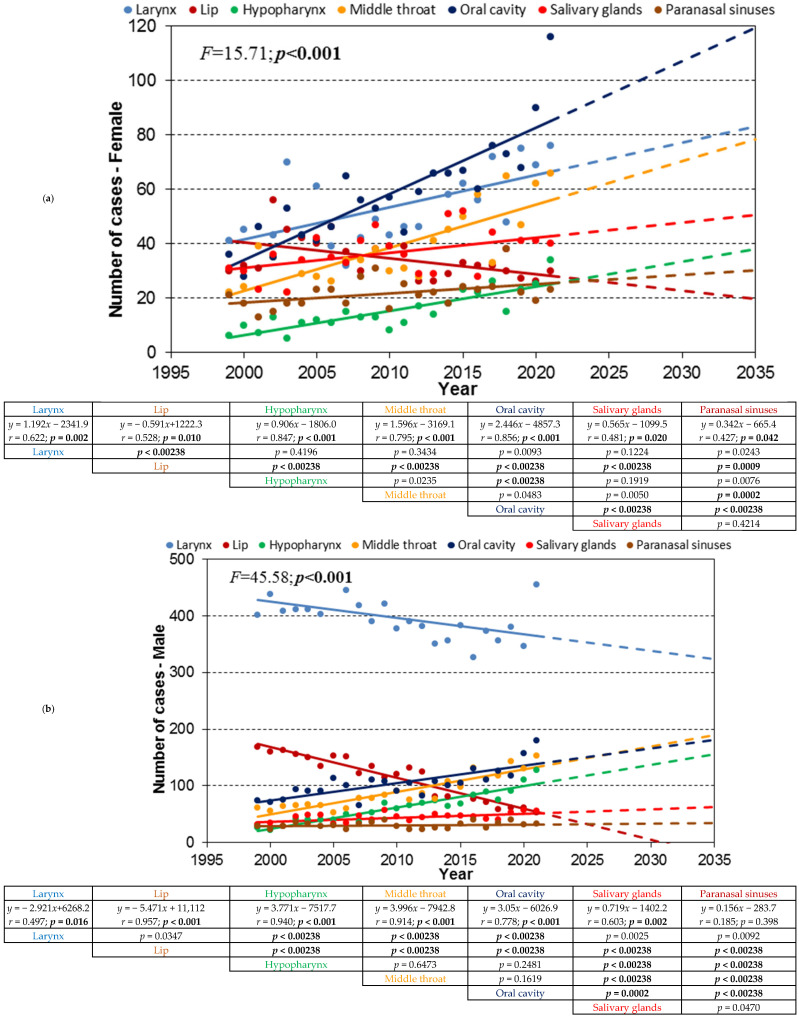
(**a**). Trend lines of the absolute number of cases of women with distinguished groups of HNCs for the age cohort 70–79; dashed lines show the extrapolation of the trend to 2035 determined in the period 1999–2021. (**b**). Trend lines of the absolute number of cases of men with distinguished groups of HNCs for the age cohort 70–79; dashed lines show the extrapolation of the trend to 2035 determined in the period 1999–2021. (**c**). Trend lines of the absolute number of cases of distinguished groups of HNCs for the age cohort 70–79; dashed lines show the extrapolation of the trend to 2035 determined in the period 1999–2021.

**Figure 4 cancers-17-00548-f004:**
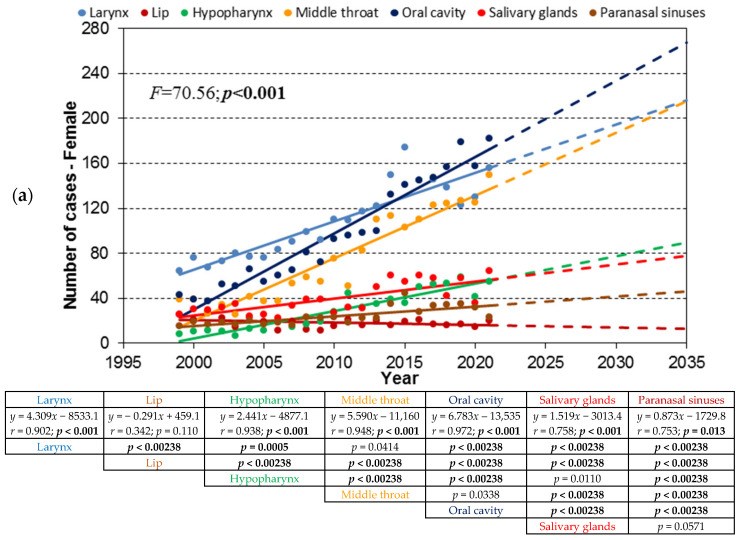
(**a**). Trend lines of the absolute number of cases of women with distinguished groups of HNCs for the age cohort 60–69; dashed lines show the extrapolation of the trend to 2035 determined in the period 1999–2021. (**b**). Trend lines of the absolute number of cases of men with distinguished groups of HNCs for the age cohort 60–69; dashed lines show the extrapolation of the trend to 2035 determined in the period 1999–2021. (**c**). Trend lines of the absolute number of cases of distinguished groups of HNCs for the age cohort 60–69; dashed lines show the extrapolation of the trend to 2035 determined in the period 1999–2021.

**Figure 5 cancers-17-00548-f005:**
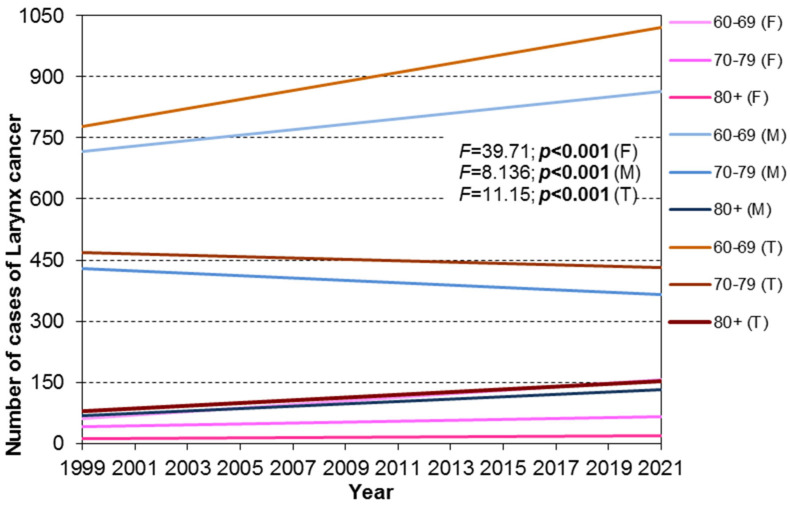
Trend lines for laryngeal cancer incidence by sex and age cohort in female (F), male (M), and total (T) populations from 1999 to 2021, stratified by age cohorts: 60–69 years, 70–79 years, and 80+ years. Parallelism tests were conducted to evaluate differences between trend lines for the analyzed age.

**Figure 6 cancers-17-00548-f006:**
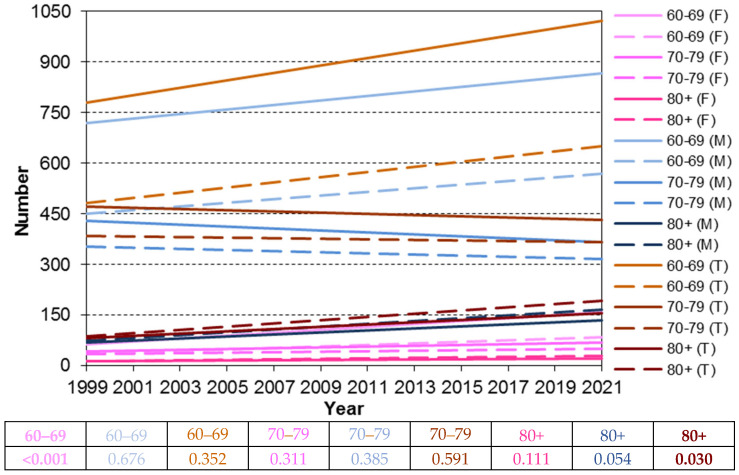
Trend lines in the number of cases and deaths (larynx cancer) for women (F), men (M), and total (T) in the period 1999–2021, considering the age cohorts 60–69, 70–79, and 80+ (results of the test of parallelism of trend lines in the number of cases and number of deaths).

**Table 1 cancers-17-00548-t001:** Results of the parallelism test for trend lines of incidence rates for selected HNSCC cancers, considering gender and age cohorts: 60–69, 70–79, and 80+.

Female				Male				Total			
Age	*b_NoC_*		*p*	Age	*b_NoC_*		*p*	Age	*b_NoC_*		*p*
Larynx											
*F* = 39.71; *p* < 0.001				*F* = 8.136; *p* < 0.001				*F* = 8.136; *p* < 0.001			
60–69 vs. 70–79 69–60 vs. 80+ 70–79 vs. 80+	4.309 4.309 1.192	1.192 0.352 0.352	<0.017 <0.017 0.022	60–69 vs. 70–79 69–60 vs. 80+ 70–79 vs. 80+	6.697 6.697 –2.921	–2.921 2.981 2.981	0.002 0.180 <0.017	60–69 vs. 70–79 69–60 vs. 80+ 70–79 vs. 80+	11.006 11.006 –1.729	–1.729 3.333 3.333	<0.017 0.017 <0.017
Lip											
*F* = 13.38; *p* < 0.001				*F* = 42.1; *p* < 0.001				*F* = 45.01; *p* < 0.001			
60–69 vs. 70–79 69–60 vs. 80+ 70–79 vs. 80+	–0.219 –0.219 –0.591	–0.591 0.767 0.767	0.138 <0.017 <0.017	60–69 vs. 70–79 69–60 vs. 80+ 70–79 vs. 80+	–3.935 –3.935 –5.471	–5.471 0.257 0.257	0.041 <0.017 <0.017	60–69 vs. 70–79 69–60 vs. 80+ 70–79 vs. 80+	–4.154 –4.154 –6.062	–6.062 1.024 1.024	0.028 <0.017 <0.017
Hypopharynx											
*F* = 51.33; *p* < 0.001				*F* = 91.37; *p* < 0.001				*F* = 104.4; *p* < 0.001			
60–69 vs. 70–79 69–60 vs. 80+ 70–79 vs. 80+	2.441 2.441 0.906	0.906 0.464 0.464	<0.017 <0.017 <0.017	60–69 vs. 70–79 69–60 vs. 80+ 70–79 vs. 80+	10.857 10.857 3.771	3.771 1.043 1.043	<0.017 <0.017 <0.017	60–69 vs. 70–79 69–60 vs. 80+ 70–79 vs. 80+	13.297 13.297 4.677	4.677 1.508 1.508	<0.017 <0.017 <0.017
Middle throat											
*F* = 86.05; *p* < 0.001				*F* = 92.99; *p* < 0.001				*F* = 121.04; *p* < 0.001			
60–69 vs. 70–79 69–60 vs. 80+ 70–79 vs. 80+	5.590 5.590 1.596	1.596 0.432 0.432	<0.017 <0.017 <0.017	60–69 vs. 70–79 69–60 vs. 80+ 70–79 vs. 80+	12.291 12.291 3.996	3.996 1.131 1.131	<0.017 <0.017 <0.017	60–69 vs. 70–79 69–60 vs. 80+ 70–79 vs. 80+	17.880 17.880 5.592	5.592 1.563 1.563	<0.017 <0.017 <0.017
Oral cavity											
*F* = 79.95; *p* < 0.001				*F* = 89.23; *p* < 0.001				*F* = 107.68; *p* < 0.001			
60–69 vs. 70–79 69–60 vs. 80+ 70–79 vs. 80+	6.783 6.783 2.446	2.446 1.980 1.980	<0.017 <0.017 0.213	60–69 vs. 70–79 69–60 vs. 80+ 70–79 vs. 80+	13.565 13.565 3.050	3.050 1.259 1.259	<0.017 <0.017 <0.017	60–69 vs. 70–79 69–60 vs. 80+ 70–79 vs. 80+	20.348 20.348 5.496	5.496 3.239 3.239	<0.017 <0.017 <0.017
Salivary glands											
*F* = 4.861; *p* = 0.011				*F* = 5.646; *p* = 0.006				*F* = 8.047; *p* < 0.001			
60–69 vs. 70–79 69–60 vs. 80+ 70–79 vs. 80+	1.519 1.519 0.565	0.565 0.821 0.821	<0.017 0.033 0.335	60–69 vs. 70–79 69–60 vs. 80+ 70–79 vs. 80+	1.795 1.795 0.719	0.719 1.407 1.407	<0.017 0.258 <0.017	60–69 vs. 70–79 69–60 vs. 80+ 70–79 vs. 80+	3.314 3.314 1.285	1.285 2.228 2.228	<0.017 0.048 0.017
Paranasal sinuses											
*F* = 3.117; *p* = 0.051				*F* = 8.401; *p* < 0.001				*F* = 9.299; *p* < 0.001			
60–69 vs. 70–79 69–60 vs. 80+ 70–79 vs. 80+	0.873 0.873 0.342	0.342 0.429 0.429	0.026 0.061 0.700	60–69 vs. 70–79 69–60 vs. 80+ 70–79 vs. 80+	1.167 1.167 0.156	0.156 0.135 0.135	<0.017 <0.017 0.935	60–69 vs. 70–79 69–60 vs. 80+ 70–79 vs. 80+	2.040 2.040 0.498	0.498 0.564 0.564	<0.017 <0.017 0.857

**Table 2 cancers-17-00548-t002:** Results of the parallelism test for trend lines of incidence vs. mortality rates for selected HNSCC cancers, considering gender and age cohorts: 60–69, 70–79, and 80+ (bNoC—the slope coefficient of the trend line of the number of cases; bNoD—the slope coefficient of the trend line of the number of deaths).

Female	Age 60–69			Age 70–79			Age 80+		
Cancer location	*b_NoC_*	*b_NoD_*	*p*	*b_NoC_*	*b_NoD_*	*p*	*b_NoC_*	*b_NoD_*	*p*
Larynx	4.309	2.313	<0.001	1.192	0.777	0.156	0.352	0.13	0.056
Lip	–0.219	–0.060	0.130	–0.591	–0.211	0.045	0.767	0.701	0.417
Hypopharynx	2.441	2.141	0.138	0.906	1.021	0.256	0.464	0.708	0.031
Middle throat	5.590	2.961	<0.001	1.596	1.050	0.048	0.432	0.668	0.117
Oral cavity	6.783	3.587	<0.001	2.446	1.593	0.019	1.980	2.320	0.120
Salivary glands	1.519	0.479	<0.001	0.565	–0.034	0.015	0.821	0.820	0.499
Paranasal sinuses	0.873	0.519	0.052	0.342	0.316	0.446	0.429	0.180	0.107
Male	Age 60–69			Age 70–79			Age 80+		
Larynx	6.697	5.373	0.338	–2.921	–1.630	0.193	2.981	4.150	0.027
Lip	–3.935	–0.578	<0.001	–5.471	–1.264	<0.001	0.257	1.127	0.034
Hypopharynx	10.857	9.929	0.220	3.771	3.836	0.453	1.043	1.486	0.023
Middle throat	12.291	9.482	0.012	3.996	3.481	0.148	1.131	1.335	0.176
Oral cavity	13.565	11.470	0.069	3.050	2.531	0.232	1.259	1.645	0.059
Salivary glands	1.795	0.813	0.003	0.719	0.661	0.407	1.407	1.251	0.204
Paranasal sinuses	1.167	0.872	0.193	0.156	0.485	0.081	0.135	0.611	0.021
Total	Age 60–69			Age 70–79			Age 80+		
Larynx	11.006	7.686	0.176	–1.729	–0.854	0.296	3.333	4.864	0.015
Lip	–4.154	–0.638	<0.001	–6.062	–1.475	<0.001	1.024	1.828	0.126
Hypopharynx	13.297	12.070	0.182	4.677	4.857	0.390	1.508	2.195	0.004
Middle throat	17.880	12.444	<0.001	5.592	4.532	0.066	1.563	2.003	0.070
Oral cavity	20.348	15.057	0.002	5.496	4.124	0.069	3.239	3.965	0.034
Salivary glands	3.314	1.292	<0.001	1.285	0.627	0.039	2.228	2.071	0.335
Paranasal sinuses	2.040	1.390	0.075	0.498	0.801	0.190	0.564	0.791	0.240

## Data Availability

Data can be obtained from the correspondence author upon request.

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
