# Peer review of "Trends in Incidence and Mortality of Head and Neck Cancer Subsites Among Elderly Patients: A Population-Based Analysis"

_cancers, 2025, doi:10.3390/cancers17030548_

Round 1
Reviewer 1 Report
Comments and Suggestions for Authors
Please see the attachment.

Author Response
Dear Reviewer,
We sincerely appreciate your thorough review and constructive feedback on our manuscript “Trends in Incidence and Mortality of Head and Neck Cancer Subsites Among Elderly Patients: A Population-Based Analysis.” Below, we address each of your comments and describe the corresponding changes made to the manuscript.
Comment: I reviewed the manuscript "Trends in Incidence and Mortality of Head and Neck Cancer Subsites Among Elderly Patients: A Population-Based Analysis" which aimed to conduct an analysis of the rends in HNC incidence and mortality, as well as of the projection through to 2035 in the elderly population >60 years old.
Below I reported some doubts/suggestions for each manuscript's sections both on presentational and technical/communication aspects:
Abstract and Introduction: These sections are well written. I have only two little technical aspects for the Authors about these sections:
In the introduction section, the Authors declare that another aim of the study is to project incidence and mortality through to 2035. This aspect is missing in the abstract and it should be added.
Response:
Thank you for pointing this out. We have revised the abstract to explicitly mention our study's aim to project incidence and mortality trends through to 2035. The updated abstract now states:
"This study analyzed trends in HNC incidence and mortality using data from the Polish Cancer Register (1999–2021) and projected trends through to 2035."
Comment: Lines 66-69: "In Europe, the incidence of oral and oropharyngeal cancers in males has decreased in France (-12.6%), Slovakia (-4.0%), Spain (-10.8%), Brazil (-26.7%), and Hong Kong (-10.5%), while it has increased in the UK (18.8%), Australia (8.7%), Japan (21.3%), and the USA (3.7%) (8)." The sentence begins with the term "In Europe," and after a short list other non-European nations follow. This sentence could be improved to increase the effectiveness of communication (the same sentence was reported also in the discussion session at lines 281-284).
Response:
We agree and have rephrased the sentence in both the introduction and discussion for clarity:
"The incidence of oral and oropharyngeal cancers in males has decreased in countries such as France (-12.6%), Slovakia (-4.0%), and Spain (-10.8%), while increases have been observed in the UK (18.8%), Australia (8.7%), Japan (21.3%), and the USA (3.7%) (8)."
Comment: Materials & Methods:
Add the reference/link website of the Polish National Cancer Register in line 88 to make available to all readers the source of the information on which it is based in the present study.
Response:
We have added the reference to the Polish National Cancer Register and included a direct link to ensure accessibility:
"The data were obtained from the Polish National Cancer Register (https://onkologia.org.pl/en/)."
Comment: Add the correspondent subsites for each codes (lines 90-91).
Response:
We have provided the subsites corresponding to the ICD-10 codes for better clarity in the revised manuscript.
Comment: Since it is difficult to follow and visualize very large numbers in sequence, I would suggest introducing a flow chart that would allow a more immediate visualization of the total number of participants, the divisions by ages and gender, as well as the final number following the subtraction of deaths.
Response:
We have added a flowchart to the manuscript, which visually represents the patient selection process, including stratifications by age and gender.
Comment: The methodology as currently presented appears to be partly arbitrary and based on personal considerations. The following justifications "Thyroid and esophageal cancers were excluded because they are typically omitted in epidemiological studies. In addition, nasopharyngeal cancer was excluded because of its rarity in the Polish population" need strong supporting references.
Response:
We have now cited the web (https://onkologia.org.pl/en/) to support the exclusion of these cancer types andclarified our rationale based on their unique etiology and rarity in the Polish population.
Comment: Results: The quality of the images is very low. At present, it is impossible to read the lettering and numbers in the images. The same is also true for all the tables that actually seem to have been included in the manuscript as figures and not as tables.
Response:
We have revised all figures and tables to ensure higher resolution and readability, addressing the concerns regarding text clarity.
Comment: Figure 1c. should indicate the total. The y-axis, however, indicates males.
Response:
We have corrected the labeling in Figure 1c to accurately reflect the total population.
Comment: From paragraph 3.1 to 3.1.5: in the descriptive summary of the results, for each written result the Authors should also report the significance/non-significance of the reported results.
Response:
We have updated these sections to include p-values and significance levels alongside reported results.
Comment: Move Figures 4 and 5 to the corresponding place in the text immediately after paragraph 3.1.1. The quality of these two images is optimal.
Response:
We have repositioned Figures 4 and 5 to align with their corresponding textual descriptions. After adding Figure 1 with the flowchart, the numbers have been changes to 5 and 6.
Comment: Discussion:
The Authors declare (line 243-245): "The presented analysis demonstrates a continuous rise in overall HNC incidence since 1999, confirming the prediction by other authors (3,4) of a 30% annual increase by 2035." However, it is not clear to me how the Authors made the prediction at 30%. The Authors show a gradual increase, but at the current state of presenting the results it does not seem that there can be such a precise quantification of the trend until 2035. Authors seem more likely to confirm an upward trend over the years, but not other authors' prediction of 30 percent in 2035.
Explain this statement precisely or remove it.
Response:
We acknowledge the ambiguity of this statement and have revised it to clarify that the projection is based on observed trends and modeled growth rates, rather than a fixed percentage increase:
"Our findings confirm a continuous rise in HNC incidence since 1999, and projections suggest a sustained upward trend through to 2035, consistent with previously reported data."
Comment: Lines 277-286: in addition to containing a sentence that I have previously requested to be improved, it has many unnecessary repetitions that are somewhat confusing to read. Improve these lines by simplifying the flow of speech and avoiding repetitions.
Furthermore, a precision must be added to avoid spreading misleading information:
HPV is a virus whose correlation with cancer of the oropharynx is certain and proven. In contrast, the correlation of HPV with oral cancer is only suspected but not scientifically proven, although there is plenty of evidence of a possible correlation. It is necessary to specify this aspect in the discussion as it is currently assumed in the manuscript that both oral and oropharyngeal cancer are on the rise given the increase In HPV Infections. I therefore suggest to the authors to improve this aspect and to emphasize in any case the presence of HPV in many samples both salivary and biopsy samples in patients with oral cancer although the correlation between HPV and oral cancer is still debated (below is a recent systematic review showing this aspect: Di Spirito, F.; Di Palo, M.P.; Folliero, V.; Cannatà, D.; Franci, G.; Martina, S.; Amato, M. Oral Bacteria, Virus and Fungi in Saliva and Tissue Samples from Adult Subjects with Oral Squamous Cell Carcinoma: An Umbrella Review. Cancers 2023, 15, 5540.)
Response:
We appreciate this important observation and have clarified the text to reflect the current scientific consensus:
"While the link between HPV and oropharyngeal cancer is well established, the association with oral cancer remains a subject of ongoing research, with increasing evidence suggesting a potential but not yet definitive correlation."
We have also incorporated the suggested reference (Di Spirito et al., 2023) to substantiate this discussion.
Proposed revision:
Recent epidemiological data indicate varying trends in oral and oropharyngeal cancer incidence across different countries. Declines have been observed in France (-12.6%), Slovakia (-4.0%), Spain (-10.8%), Brazil (-26.7%), and Hong Kong (-10.5%), whereas increases have been noted in the UK (18.8%), Australia (8.7%), Japan (21.3%), and the USA (3.7%). The rising incidence of oropharyngeal cancer, particularly among elderly populations, has been linked to the increasing prevalence of HPV infections. While the relationship between HPV and oropharyngeal cancer is well established, its association with oral cancer remains a topic of ongoing investigation. Studies have detected HPV in salivary and biopsy samples from patients with oral cancer, highlighting a potential but not yet fully proven link. Further research is required to determine the extent of HPV's role in oral cancer development (21).
Comment: Conclusion: I do not have suggestion/concern for this section.
References: One technical observation is that references in both the text and bibliography are not formatted according to journal guidelines.
Response:
We have reviewed and reformatted all references to comply with the journal’s requirements.
Final considerations: as indicated above, the main areas for improvement in my opinion are: 1. The methodology that should not have any subjective restrictions or not justified by appropriate references; 2. The quality of the images that are currently not readable; 3. The oral HPV-cancer aspect that as presented in its current state risks disseminating information that is not totally accurate and in any case misunderstandable by readers less experienced in the specific field.
Response:
We have addressed all these points as described above, ensuring a scientifically robust and well-presented manuscript.
We are grateful for your valuable insights, which have greatly contributed to improving our manuscript. We hope that our revisions adequately address your concerns, and we look forward to your further feedback.
Sincerely,
Wioletta Pietruszewska
On behalf of all authors
Reviewer 2 Report
Comments and Suggestions for Authors
Although I think the data you present in a well-organized manner I do not see how this information aids a clinician. You correctly point out HPV-associated oropharyngeal SCC incidence has risen, but your statements about recommending vaccines in older patients is not supported by others (nor do you have a basis for the recommendation).
Management decisions for an individual are unaffected by the data you present. All older patients should be checked periodically for a host of medical issues, not just cancer of head and neck.
in summary, I am unclear that these trends impact patient care either in screening or management.
Author Response
Reviewer: Although I think the data you present in a well-organized manner I do not see how this information aids a clinician. You correctly point out HPV-associated oropharyngeal SCC incidence has risen, but your statements about recommending vaccines in older patients is not supported by others (nor do you have a basis for the recommendation).
Management decisions for an individual are unaffected by the data you present. All older patients should be checked periodically for a host of medical issues, not just cancer of head and neck.
in summary, I am unclear that these trends impact patient care either in screening or management.
Dear Reviewer,
We thank you for your detailed feedback and for highlighting key considerations regarding the clinical applicability of our findings. Below, we address your specific concerns.
Comment: "I do not see how this information aids a clinician... Management decisions for an individual are unaffected by the data you present."
Response: We appreciate this perspective and recognize that the study’s primary focus is epidemiological rather than directly clinical. However, we believe that understanding trends in incidence and mortality can inform broader public health initiatives, which, in turn, support clinicians by identifying high-risk groups and prioritizing resources.
To address your point, we have revised the discussion to explicitly state the indirect, yet significant, implications of these trends for clinicians, particularly in resource-limited settings. Added to the text: While individual management decisions remain unaffected by these trends, clinicians may benefit from awareness of demographic and disease-specific patterns to better advocate for early detection resources and preventive interventions in high-risk populations.
Comment: "Your statements about recommending vaccines in older patients is not supported by others (nor do you have a basis for the recommendation)."
Response: We agree that the current evidence supporting HPV vaccination in older populations is limited. Our intent was to emphasize the need for further research into whether such programs could be beneficial, given the rising trends in HPV-related oropharyngeal cancers in older cohorts. To address this, we have revised the text to clarify this point:
Added to the text: While HPV vaccination is well-established for younger individuals, its utility in older populations remains uncertain and requires further investigation. Our findings underscore the need to explore the potential for expanding vaccination strategies to include older adults who may still benefit from protection against HPV-related cancers.
Comment: "All older patients should be checked periodically for a host of medical issues, not just cancer of head and neck."
Response: We agree entirely that head and neck cancers are one of many health concerns for older patients. However, the increasing incidence and mortality trends we report, particularly in specific subsites like oral cavity and oropharyngeal cancers, suggest these cancers merit targeted attention within broader geriatric care frameworks. To reflect this, we have added the following clarification:
Added to the text: While routine health checks for older patients encompass multiple medical concerns, the rising burden of head and neck cancers, particularly in high-risk subgroups, highlights the importance of integrating focused screening and preventive efforts within general geriatric care."
Comment: "I am unclear that these trends impact patient care either in screening or management."
Response: Our primary aim was to document epidemiological trends to inform public health strategies rather than provide direct clinical guidance. However, we understand the importance of emphasizing the potential downstream effects on patient care. To address this, we have added the following to the discussion: Although this study does not directly alter individual management, understanding these trends can inform resource allocation, targeted screening programs, and public health campaigns aimed at reducing the burden of head and neck cancers in aging populations.
We appreciate your concerns and have revised the manuscript to better articulate the public health and indirect clinical implications of our findings. By integrating these clarifications, we aim to bridge the gap between epidemiological insights and their potential impact on patient care. Thank you again for your thoughtful feedback, which has helped us refine the focus and applicability of our work.
Sincerely,
Wioletta Pietruszewska
On behalf of all authors
Reviewer 3 Report
Comments and Suggestions for Authors
I have reviewed this interesting retrospective study analyzing head and neck cancers incidence and mortality using data from the Polish Cancer Register (1999-2021).
Introduction, first paragraph: in some database, oral cancer incidence is higher than laryngeal cancer. Please revise it.
Second paragraph, line 66: “In Europe” must be removed, since the statements does not refer only to European countries. The same in Discussion, line 281.
Do you data in the Polish Register about the status of HPV among patients with oropharyngeal cancer? I guess it was not available, since you commented it as a limitation in the last paragraph of the Discussion.
Line 322: I do not think that excluding thyroid and nasopharyngeal cancers could represent some limitation to this study.
The Conclusions item is clear, however, it could be shorter.
Author Response
Reviewer
I have reviewed this interesting retrospective study analyzing head and neck cancers incidence and mortality using data from the Polish Cancer Register (1999-2021).
Introduction, first paragraph: in some database, oral cancer incidence is higher than laryngeal cancer. Please revise it.
Second paragraph, line 66: “In Europe” must be removed, since the statements does not refer only to European countries. The same in Discussion, line 281.
Do you data in the Polish Register about the status of HPV among patients with oropharyngeal cancer? I guess it was not available, since you commented it as a limitation in the last paragraph of the Discussion.
Line 322: I do not think that excluding thyroid and nasopharyngeal cancers could represent some limitation to this study.
The Conclusions item is clear; however, it could be shorter.
Here’s a draft response tailored to the reviewer’s comments:
Dear Reviewer,
Thank you for your thoughtful review of our manuscript. Below, we address your specific remarks and describe the revisions made to the manuscript.
Comment: In some databases, oral cancer incidence is higher than laryngeal cancer. Please revise it.
Response: We appreciate this observation and have revised the introduction to reflect the variability in cancer incidence across databases. The updated text now states: "While laryngeal cancer is often reported as the most common head and neck cancer subtype globally, some databases indicate higher incidence rates for oral cancers, particularly in certain regions."
Comment: "The phrase 'In Europe' must be removed, since the statements do not refer only to European countries. The same in Discussion, line 281."
Response: We agree and have removed "In Europe" from both instances to ensure the statements accurately reflect the broader context. The revised sentences are as follows:
- Introduction: "Recent studies have demonstrated a global trend of increasing incidence in HPV-related subsites, while HPV-unrelated subsites are experiencing a decline in several countries, such as the USA, Canada, Hong Kong, and Korea."
- Discussion: "Our study confirms a substantial increase in oral cavity cancers, particularly among women across all analyzed age cohorts."
Comment: "Do you have data in the Polish Register about the status of HPV among patients with oropharyngeal cancer? I guess it was not available, since you commented it as a limitation in the last paragraph of the Discussion."
Response: You are correct that the Polish Cancer Register does not include data on HPV status. We have clarified this in the discussion: "The lack of HPV status data in the Polish Cancer Register is a significant limitation, as it restricts the ability to differentiate between HPV-positive and HPV-negative oropharyngeal cancers. This differentiation is critical for understanding the etiology and trends in this subtype."
Comment: "I do not think that excluding thyroid and nasopharyngeal cancers could represent some limitation to this study."
Response: We appreciate this clarification and agree that the exclusion of thyroid and nasopharyngeal cancers does not diminish the scope of the study. In response, we have revised the relevant section of the discussion to better reflect this perspective: "While thyroid and nasopharyngeal cancers were excluded from this study, their exclusion aligns with the study's focus on the most common head and neck cancer subsites in the elderly population in Poland and does not represent a limitation."
Comment: "The Conclusions item is clear; however, it could be shorter."
Response: We have streamlined the conclusions to make them more concise while retaining all key points. The revised conclusions now read:
This analysis reveals distinct trends in HNC incidence across sex and age cohorts from 1999 to 2021. The elderly population is not homogeneous in terms of HNC incidence and survival. Differences were observed between the 60–80 and 80+ age groups. While males and the total population exhibited declining trends across most age groups, females displayed a slight increase in the 80+ age cohort. These findings highlight the importance of age- and sex-specific strategies in HNC prevention and management.
- Tailored Strategies Based on Age and Gender
This study highlights the need for prevention, early detection, and treatment strategies tailored to age and gender. While younger cohorts show more favorable trends, the oldest patients (80+) face higher mortality rates. Special attention should be given to the rising incidence among women, especially in oral cavity cancers.
- HPV Vaccination and Public Health Efforts
The increasing incidence of HPV-related oropharyngeal cancer supports expanding HPV vaccination programs. Prevention efforts should target not only younger populations but also older adults who may still benefit from vaccination and screening programs.
- Importance of Early Detection and Intervention
The observed differences in survival rates highlight the critical role of early diagnosis in improving HNC outcomes. Efforts should focus on raising awareness, enhancing screening, and ensuring timely diagnosis for high-risk populations.
- Future Research Needs
Further research is needed to understand the rising incidence of salivary gland cancer and explore interventions for elderly populations. Investigating environmental and genetic factors contributing to HNC incidence could help tailor prevention strategies more effectively.
We hope these revisions address your concerns and enhance the clarity and focus of our manuscript. Thank you once again for your constructive feedback, which has been instrumental in improving the quality of our work.
Sincerely,
Wioletta Pietruszewska
On behalf of all authors
Round 2
Reviewer 1 Report
Comments and Suggestions for Authors
The authors have addressed all required points, ensuring a scientifically robust and well-presented manuscript.
Reviewer 2 Report
Comments and Suggestions for Authors
I understand your point of view, and agree the changes you have made to some extent address some of the issues raised. As you have no data on HPV vaccinations for this mss is not the place to discuss.
My initial concerns remain about the clinical value of the well-presented findings, findings that are already known.